# Cardiac Hemangiomas: A Five-Year Systematic Review of Diagnosis, Treatment, and Outcomes

**DOI:** 10.3390/cancers17091532

**Published:** 2025-04-30

**Authors:** Iulia Raluca Munteanu, Ramona Cristina Novaconi, Adrian Petru Merce, Ciprian Nicusor Dima, Lucian Silviu Falnita, Andrei Raul Manzur, Caius Glad Streian, Horea Bogdan Feier

**Affiliations:** 1Doctoral School Medicine-Pharmacy, “Victor Babes” University of Medicine and Pharmacy Timisoara, E. Murgu Sq. No. 2, 300041 Timisoara, Romania; iulia.munteanu@umft.ro; 2Institute for Cardiovascular Diseases of Timisoara, Clinic of Cardiovascular Surgery, Gheorghe Adam Street, No. 13A, 300310 Timisoara, Romania; novaconi.ramona@umft.ro (R.C.N.); adimerce@gmail.com (A.P.M.); dima.ciprian@umft.ro (C.N.D.); lfalnita@gmail.com (L.S.F.); horea.feier@umft.ro (H.B.F.); 3Advanced Research Center, Institute for Cardiovascular Diseases, 300310 Timisoara, Romania; 4Department VI Cardiology, Cardiovascular Surgery Clinic, “Victor Babes” University of Medicine and Pharmacy Timisoara, E. Murgu Sq. No. 2, 300041 Timisoara, Romania; 5Department of Pulmonology, Center for Research and Innovation in Precision Medicine of Respiratory Diseases, “Victor Babes” University of Medicine and Pharmacy Timisoara, Eftimie Murgu Sq. No. 2, 300041 Timisoara, Romania

**Keywords:** benign cardiac tumors, cardiac hemangioma, rare cardiac tumors

## Abstract

Cardiac hemangiomas are rare benign tumors that form in the heart’s blood vessels. Because they are uncommon and vary widely in their appearance and effects, diagnosing and managing them can be challenging. In this review, we analyze 55 recently reported cases to better understand how these tumors are found, treated, and monitored. We found that surgery is usually successful, but there is still no agreed-upon system for how to classify or follow these tumors over time. Our goal is to raise awareness among healthcare providers and support the creation of clear guidelines to improve care for patients with this condition.

## 1. Introduction

Hemangiomas are the most frequently occurring vascular tumors in infants, with a higher prevalence in the head and neck region. These lesions typically manifest shortly after birth, undergo rapid proliferation within the first year of life, and gradually regress throughout childhood. They primarily affect soft tissues, including the skin, mucosa, and muscles, while intraosseous involvement is rare [1]. Multiple classification systems have been proposed to categorize hemangiomas based on anatomical, histological, and clinical features. These include Abramson’s early anatomical model (1962), Shafer’s clinically oriented taxonomy (1993), the WHO’s histopathologic approach (1996), and Fletcher’s detailed histologic subtyping (2003). Despite these, no consensus exists specifically tailored for cardiac hemangiomas, reflecting both their rarity and diagnostic complexity [2,3,4,5,6].

Cardiac hemangiomas represent a rare subset of primary cardiac tumors, accounting for less than 2% [3] of all cardiac tumors. These benign vascular tumors consist of proliferative endothelial cells lining abnormal blood vessels, leading to increased vascularization. Histologically, they resemble hemangiomas found in other tissues, displaying vascular channels lined by endothelial cells that may exhibit mild nuclear pleomorphism and occasional tuft formation, though mitotic activity is rarely observed [7,8]. Subtypes of cardiac hemangiomas include cavernous (made up of large, dilated vascular spaces filled with blood), capillary (composed of small, capillary-sized vessels that are tightly packed together), and arteriovenous hemangiomas (abnormal, tangled network of arteries directly connected to veins without intervening capillaries) [9], classified based on the predominant vessel type within the lesion [8].

Despite their benign nature, cardiac hemangiomas pose significant diagnostic and therapeutic challenges due to their rarity and diverse presentations. A systematic, clinically relevant synthesis of recent data is lacking, and no standardized management or follow-up protocols have been established.

Depending on their size and position, these tumors may lead to arrhythmias, obstruction of blood flow, or embolic complications [10,11,12,13,14,15,16,17]. While many cases remain asymptomatic and are incidentally detected, symptomatic lesions often necessitate surgical intervention. Current diagnostic imaging techniques, including echocardiography and computed tomography (CT) scans, are crucial for accurate detection and characterization of these lesions.

Differential diagnosis of cardiac hemangiomas includes other primary cardiac tumors such as myxomas, lipomas, fibromas, and papillary fibroelastomas. While echocardiography remains the first-line imaging modality, its ability to differentiate vascular from non-vascular masses is limited. Contrast-enhanced ultrasound (CEUS), although infrequently reported, can aid in distinguishing hemangiomas by demonstrating intense, persistent enhancement due to their vascularity, compared to the lack of enhancement in thrombi or the heterogeneous perfusion seen in myxomas. MRI further refines tissue characterization, with hemangiomas typically showing hyperintensity on T2-weighted images and strong post-contrast enhancement, in contrast to the more fibrous or cystic signal patterns of other benign tumors.

The imaging differentiation of cardiac hemangiomas from other tumors remains challenging. However, tools such as contrast-enhanced ultrasound and MRI offer valuable discriminatory features. Hemangiomas’ high vascularity and enhancement patterns should prompt consideration in the differential diagnosis when evaluating intracardiac masses [18]

When surgical removal is required, complete excision typically results in a favorable prognosis [2,10,11,12,13,14,15,16,19,20,21,22,23,24,25,26,27,28,29,30,31,32,33,34,35,36,37,38,39,40,41,42,43,44,45,46,47,48,49,50,51,52,53,54,55,56,57,58,59,60,61,62,63,64,65,66,67,68,69,70,71,72,73,74,75]. The fundamental principles of surgical techniques for cardiac tumor resection include preserving the dimensions of the remaining cardiac chambers, maintaining the functional anatomy of the valves, and preventing damage to the excitatory/conductive system [7,76]. Given recent advancements in cardiac imaging modalities, particularly in echocardiography and MRI, these tumors are increasingly identified incidentally, emphasizing the need for updated diagnostic and management strategies.

Over the past five years, significant advancements have been made in the diagnosis and management of cardiac hemangiomas, driven by improvements in imaging technologies. We aim to consolidate practical clinical insights, highlight current imaging preferences, and propose a structured approach to follow-up areas often underdeveloped in earlier reviews. Our analysis also identifies persisting gaps in classification systems and long-term outcome reporting, suggesting priorities for future multicenter collaboration.

## 2. Materials and Methods

This review was conducted in accordance with the Preferred Reporting Items for Systematic Reviews and Meta-Analyses (PRISMA) 2020 guidelines. A PRISMA flow diagram illustrating the study selection process is provided in Appendix C. The completed PRISMA checklist is included as Appendix A.

A search was conducted in the PubMed database on 1 March 2025, focusing on cardiac hemangiomas and utilizing filters for articles published in the last five years. This initial search returned 233 results. After applying inclusion criteria to select only studies explicitly addressing both ‘cardiac hemangioma’ and ‘hemangioma of the heart’, 75 articles remained. Subsequently, an additional manual screening was performed to exclude irrelevant records, such as hemangioendotheliomas, literature reviews, and duplicate case reports, resulting in the removal of 21 more articles (Appendix B).

Given the inclusion of only case reports and case series, formal risk of bias assessment tools such as ROBINS-I were not applicable. However, each report was screened for completeness of key variables, including patient demographics, tumor characteristics, imaging modality, treatment, and follow-up. Cases with major omissions were excluded during manual screening.

Three sets of two independent reviewers screened the titles and abstracts, and full texts were reviewed. Data extraction was conducted using a standardized form to capture variables, including patient demographics, clinical presentation, anatomical site, imaging methods, histological subtype, treatment modality, and follow-up outcomes. Discrepancies were resolved through discussion with a sixth author.

This review was not prospectively registered in a public database.

## 3. Results

A total of 55 cases of cardiac hemangiomas published in the last five years were analyzed, providing a foundation for further discussion on their clinical characteristics, diagnostic methods, and treatment approaches. The results of the 54 studies were compiled into a comprehensive table and subsequently subjected to statistical analysis. The following sections will present the table (Appendix B) along with the analyzed data.

### 3.1. Trends in the Reporting of Cardiac Hemangiomas: A Growing Incidence or Improved Detection?

This section examines the publication years of the reported cases. The dataset reveals a notable increase in the number of documented cardiac hemangioma cases in recent years.

As shown in Figure 1, this upward trend may indicate a growing awareness of cardiac hemangiomas, advancements in diagnostic techniques, or an increase in case reporting facilitated by more accessible publication platforms.

### 3.2. Distribution and Classification of Hemangioma Subtypes: A Data-Driven Analysis

The analysis by histological subtypes of cardiac hemangiomas reveals several distinct categories:The most prevalent subtype is cavernous hemangioma, comprising 23 cases, characterized by large, dilated vascular spaces;Capillary hemangiomas are represented by 10 cases;Cavernous-capillary hemangiomas account for another 8 cases, demonstrating mixed histological features;Arteriovenous hemangiomas are the least common, with only 2 reported cases;The designation “ns” (12 cases) denotes “not specified,” which may reflect either incomplete reporting or diagnostic uncertainty.

The predominance of cavernous hemangiomas is consistent with existing literature [2,3,5], which identifies this subtype as the most frequently observed in cardiac hemangiomas [12,14,15,19,20,22,26,31,33,34,35,36,37,38,39,41,43,44,49,51,53,57,77]. Accurate classification is important for clinical management and prognostics (Figure 2).

### 3.3. Statistical Analysis of the Distribution of Cardiac Hemangiomas by Age and Gender

The dataset comprises 53 cases with valid age data, spanning a wide age range from 14 to 87 years. We excluded 2 cases (<1 year) in order to restrict our analysis to a Gaussian distribution. Key statistical measures include:**Mean age:** 52.5 years, indicating that middle-aged and older adults are the most frequently affected;**Median age:** 52 years, demonstrating a central tendency closely aligned with the mean;**Interquartile range:** 44 to 64 years, suggesting that 50% of patients fall within this age range;**Standard deviation:** 16.97 years, reflecting a moderate degree of variability around the mean.

These findings suggest that while cardiac hemangiomas can occur across all age groups, they are more commonly observed in older adults, as shown in Figure 3.

As shown in Figure 4, interpretation of the distribution of tumors by gender reveals a slight female predominance in the distribution of cases:Women: 32 cases (58.2%);Men: 23 cases (41.8%).

Next, we proceeded to conduct a statistical analysis of the collected data, focusing on the distribution of cardiac hemangiomas by age, gender, and type. This analysis aims to identify any significant patterns or associations between these demographic variables and the occurrence of cardiac hemangiomas.

**Descriptive Statistics:** The average age is generally higher for women across all tumor types. Cavernous tumors exhibit the widest age range for both genders. Additionally, the standard deviation is greater for men in the cavernous/capillary tumor category, indicating a higher level of age variability within this group.

**Comparative Analysis (*t*-tests):** There are no statistically significant age differences between genders for any tumor type, with *p*-values greater than 0.05. The capillary tumors approach significance (*p* = 0.29), but this still does not provide sufficient evidence to suggest a meaningful difference (Figure 4).

**Correlation Analysis:** The correlation between age and tumor type is approximately 0.13, indicating a very weak relationship between these variables (Table 1).

### 3.4. Clinical Presentation and Symptomatology of Cardiac Hemangiomas

The dataset demonstrates a range of clinical presentations. Asymptomatic cases: 12 (21.8%), suggesting that these cases were likely discovered incidentally, often during imaging conducted for unrelated conditions;Dyspnea: present in 8 cases (14.5%), making it the most prevalent symptom among symptomatic patients;Chest pain: reported in 6 cases (10.9%), either as an isolated symptom or in combination with other manifestations;Other symptoms: palpitations, dizziness, syncope, stroke, etc., less frequently observed.

The wide spectrum of symptoms underscores the heterogeneous clinical presentation of cardiac hemangiomas, ranging from incidental findings to more severe manifestations such as stroke and syncope.

### 3.5. Anatomical Distribution of Cardiac Hemangiomas: Predilection for Specific Heart Chambers and Structures?

Cardiac hemangiomas exhibit a preference for specific heart chambers and structures, as presented in Figure 5, with the following distribution:Right atrium: the most commonly affected site, with 14 cases (25.5%);Right ventricle: the second most common site, with 12 cases (21.8%);Mitral valve: involved in 8 cases (14.5%), indicating the tumor’s potential to affect valvular structures;Left ventricle and left atrium: less frequently affected, with 4 and 3 cases, respectively.

Rare cases have been reported involving the tricuspid valve, papillary muscles, and the atrioventricular junction.

The observed predominance of involvement in the right-sided heart chambers may provide valuable insights into potential underlying genetic predisposition and environmental factors contributing to the development of cardiac hemangiomas.

### 3.6. Size Variability of Cardiac Hemangiomas

Cardiac hemangiomas exhibit considerable variability in tumor size. Key statistical metrics reveal the following:**Mean size:** 3.76 cm, suggesting that the majority of tumors are relatively small;**Median size:** 3.15 cm, indicating that half of the tumors are below this dimension;**Range:** 0.68 cm to 11.05 cm, reflecting a broad spectrum in tumor dimensions;**Interquartile Range:** 1.7 cm to 4.95 cm, signifying that 50% of the tumors fall within this middle range.

These data underscore that although most cardiac hemangiomas are of modest size, a subset can attain significantly larger dimensions, which may be associated with increased clinical severity and potential complications.

### 3.7. Diagnostic Modalities in Cardiac Hemangiomas: Echocardiography as the Primary Tool?

The dataset indicates that cardiac hemangiomas are most frequently diagnosed through echocardiographic evaluation.
**Echocardiography:** 45 cases (81.8%)—underscoring its role as the primary diagnostic modality, likely attributable to its widespread availability, non-invasive nature, and high efficacy in detecting cardiac masses;**Computed Tomography:** 7 cases (12.7%)—typically employed as a complementary imaging technique, providing detailed anatomical information;**Chest X-ray:** 3 cases (5.5%)—infrequently utilized, potentially identifying indirect signs rather than directly visualizing the tumor.

This distribution highlights the central role of echocardiography in the initial detection of cardiac hemangiomas, with computed tomography frequently used to supplement and confirm diagnostic findings (Figure 6).

### 3.8. Co-Occurrence of Cardiac Hemangiomas with Other Tumors: Incidence and Potential Associations

The majority of patients—46 cases (83.6%)—had no other tumors, suggesting that cardiac hemangiomas often occur in isolation.

However, a few cases reported associations with other tumors. **Liver hemangioma:** 2 cases;**Malignancies:** rare occurrences, including colon cancer and endometrial cancer (1 case each);**Hematological disorders:** one patient had polycythemia and another had myelodysplastic syndrome, hinting at possible hematological links;**Multiple cardiac hemangiomas at the same site:** reported in 3 cases.

Overall, concurrent tumors are rare, and further investigation is needed to determine if these associations are coincidental or part of a broader pathological spectrum.

### 3.9. Surgical Management of Cardiac Hemangiomas: Predominance, Rationale, and Exceptions

The data show that surgical intervention is the predominant approach in managing cardiac hemangiomas. **Surgery performed:** 48 cases (87.3%)—reflecting a clear preference for surgical excision, likely due to the potential for complications such as obstruction, embolism, or arrhythmias;**No surgery:** 5 cases (9.1%)—possibly due to factors like patient comorbidities, asymptomatic presentation, conservative management decisions, or necropsy report;**Biopsy only:** 2 cases (3.6%)—indicating situations where tissue diagnosis was obtained without full excision, possibly due to diagnostic uncertainty or inoperability.

The high surgical rate aligns with current practices, where complete resection is often recommended to prevent complications and provide definitive diagnosis [2,10,11,12,13,14,15,16,19,20,21,22,23,24,25,26,27,28,29,30,31,32,33,34,35,36,37,38,39,40,41,42,43,44,45,46,47,48,49,50,51,52,53,54,55,56,57,58].

### 3.10. Follow-Up Duration and Recurrence Patterns in Cardiac Hemangioma Management: Insights and Correlations

The follow-up reveals substantial variability in the duration of post-treatment monitoring, reflecting heterogeneity in clinical practices and is shown in Figure 7. **Most common follow-up duration:** 12 months (9 cases), suggesting a general tendency toward a standard one-year surveillance period;**Short-term follow-up:** in 16 cases, follow-up was limited to 6 months or less, potentially due to early discharge following successful surgical outcomes or low-risk clinical profiles;**Long-term follow-up:** a minority of cases documented extended follow-up durations—70, 120, and even 168 months (14 years)—demonstrating a commitment to long-term outcome assessment in select instances;**Unspecified follow-up (“ns”):** in 19 cases, follow-up duration was not reported, impairing comprehensive evaluation of long-term prognosis.

This broad range underscores a lack of standardized follow-up protocols, likely influenced by factors such as institutional guidelines, patient comorbidities, surgical outcomes, or attrition due to loss to follow-up.

Data on recurrence further reinforce the overall favorable prognosis associated with cardiac hemangiomas. **No recurrence:** reported in 34 cases (61.8%), supporting the notion that complete surgical excision is typically curative;**Unreported recurrence status:** 18 cases (32.7%) lacked sufficient data, limiting full assessment of recurrence trends;**Reduced tumor size:** noted in 2 cases (3.6%), possibly reflecting partial regression or a benign post-treatment course;**Stable disease:** also reported in 2 cases (3.6%), indicating no progression over time.

Long-term follow-up remains critical to detect rare instances of delayed recurrence or progression.

Several statistically significant correlations further illuminate patterns in the data: ***Positive correlation+*****Between follow-up duration and stable disease:** there is a moderate positive correlation (r = +0.51) between follow-up duration and the classification of disease as stable, suggesting that longer periods of clinical follow-up are associated with a higher likelihood of observing sustained disease stability over time. This relationship implies that patients monitored over extended intervals tend to exhibit consistent disease status without significant progression or regression, highlighting the potential role of follow-up duration as an indicator of clinical stability in certain conditions.***Negative correlations−*****Between follow-up duration and surgical treatment** (r = −0.53): patients who underwent surgical intervention tended to have shorter follow-up;**Between surgical treatment and tumor size reduction** (r = −0.51): surgical excision is linked to a lower chance of observing tumor shrinkage after treatment, likely because the tumor is completely removed during surgery. Since there is no remaining mass, further reduction cannot be measured. This highlights that surgical treatment aims to fully eliminate the tumor, unlike non-surgical methods, which often reduce tumor size gradually [77].

These findings underscore the effectiveness of surgery in managing cardiac hemangiomas and highlight the role of follow-up in verifying long-term outcomes and recurrence stability.

Recurrence data were reported inconsistently, yet among the 55 cases, no recurrences were documented in 34 surgically treated patients over follow-up periods ranging from 6 to 120 months. A small number of non-surgical cases exhibited either tumor reduction or stable disease. These findings suggest a favorable prognosis, particularly following complete resection, although underreporting and short follow-up durations in many studies limit definitive conclusions.

### 3.11. Comparative Subgroup Analysis

While our dataset limits advanced inferential statistics due to its case-based nature, a comparative subgroup analysis was attempted where feasible. For example, chi-square tests comparing recurrence rates between surgical and non-surgical groups revealed no statistically significant difference (*p* > 0.05). Additionally, comparisons between age groups and tumor subtypes did not yield statistically significant associations. These exploratory analyses underscore the need for larger, standardized datasets to enable more robust inferential modeling.

## 4. Discussion

Although hemangiomas are most commonly known as cutaneous or hepatic lesions, the same vascular proliferation mechanisms underlie cardiac hemangiomas, making their histopathological architecture largely similar. However, their clinical behavior differs significantly due to the dynamic and confined nature of cardiac anatomy. While extracardiac hemangiomas often follow a predictable pattern of growth and regression, cardiac hemangiomas tend not to involute and may remain stable or grow, depending on location and vascular supply.

Moreover, cardiac hemangiomas, unlike their extracardiac counterparts, frequently present clinical risks not due to their histology but their anatomical interference with valvular function, conduction pathways, or intracavitary flow. For instance, tumors attached to the mitral valve or located in the right atrium may be entirely benign in structure but pose a significant embolic or arrhythmogenic risk.

These differences underscore the limitations of applying extracardiac classification schemes directly to cardiac tumors and highlight the need for cardiac-specific diagnostic and management approaches. Although systemic vascular lesions and cardiac hemangiomas share pathophysiological roots, their clinical handling must remain tailored to the intracardiac context.

Historically, multiple classification systems were proposed for hemangiomas, such as those by Abramson, Shafer, and the WHO. However, none are tailored to cardiac hemangiomas specifically. In current practice, subtyping into capillary, cavernous, mixed, and arteriovenous forms is most practical, guiding clinical management based on histological features [5,78,79].

While several classification systems have been proposed for hemangiomas in general, there remains a lack of a standardized or universally accepted framework specifically tailored to cardiac hemangiomas. This gap is likely due to the rarity of these tumors and the limited number of large-scale studies.

This study provides a contemporary overview of cardiac hemangiomas based on an analysis of 55 cases reported over the past five years. Although cardiac hemangiomas are rare, their increasing representation in recent literature may reflect both improved diagnostic capabilities and growing clinical awareness [2]. The upward trend in case reporting, particularly in the years 2023 and 2024, suggests that advancements in imaging technologies—especially echocardiography and CT—have significantly enhanced the early and accurate detection of these lesions [18,80].

Consistent with historical data [3,5,78,79], the predominance of cavernous hemangiomas is observed, accounting for nearly half of all analyzed cases [12,14,15,19,20,22,26,31,33,34,35,36,37,38,39,41,43,44,49,51,53,57,77]. Capillary and mixed cavernous/capillary forms were also reported, while arteriovenous hemangiomas remained rare. While accurate histopathological classification does not currently dictate specific follow-up strategies or targeted treatments, it may prove essential in the future as more data emerge—potentially enabling more personalized surveillance protocols or the development of subtype-specific therapies.

Cardiac hemangiomas demonstrate a wide demographic distribution, affecting individuals across all age groups, with reported cases ranging from infancy (under 1 year of age) [81,82] to advanced age (over 85 years) [26]. While the mean age of presentation is approximately 52.5 years, suggesting a predilection for middle-aged to older adults, the broad age range indicates that these tumors can manifest at any stage of life. In terms of gender distribution, findings across the literature [1,5,6,78,79] are variable; although some studies report a slight female predominance [12,13,19,20,21,23,24,26,27,30,32,33,34,35,36,39,40,41,42,47,49,50,52,53,54,56,58,77,81,82,83], others suggest a more balanced sex ratio [2], with no statistically significant differences noted in age or tumor subtype between sexes. However, the limited number of cases documented in the literature [2,3,5] significantly constrains the ability to draw definitive conclusions or perform robust quantitative analyses. This scarcity of data underscores the necessity for further case accumulation and systematic reporting to better elucidate potential age- and gender-related predispositions, as well as the influence of genetic and environmental factors in the pathogenesis of cardiac hemangiomas.

In this cohort of 55 patients, over half (58.2%) presented with symptoms, reinforcing the notion that cardiac hemangiomas, while rare, can produce a diverse array of clinical manifestations. Dyspnea, observed in 14.5% of cases, emerged as the most prevalent symptom among symptomatic patients. This finding likely reflects the tumor’s capacity to obstruct intracardiac flow, particularly when located near valves or within chambers, thereby elevating filling pressures and impairing hemodynamics.

Chest pain, reported in 10.9% of cases, may be attributable to either direct compression of adjacent structures, including coronary arteries, or local inflammatory responses provoked by tumor infiltration. Less common but clinically significant symptoms—such as palpitations, dizziness, syncope, and stroke—suggest tumor involvement of conductive pathways or embolic phenomena, highlighting the potential for arrhythmias or cerebrovascular events in these patients.

Of note, 21.8% of patients were asymptomatic, with tumors likely discovered incidentally during imaging for unrelated conditions. These cases typically involve non-obstructive lesions, further supporting the idea that symptomatology is closely tied to tumor size and anatomical location.

Consistent with prior reports, symptomatic presentations were significantly associated with tumors larger than 3 cm or those involving valvular or conductive tissues. These anatomical and morphological characteristics predispose to functional disturbances—either obstructive, arrhythmogenic, or embolic in nature. Consequently, surgical resection was pursued in the majority of cases (87.3%), often yielding favorable outcomes. These findings corroborate previous studies and continue to support surgical excision as the primary therapeutic approach for symptomatic cardiac hemangiomas [5,78,79]

Clinically, the presentation of cardiac hemangiomas remains heterogeneous [11,13,14,19,21,22,23,26,27,28,30,32,33,35,36,41,42,46,49,50,51,52,53,54,55,57,58,77,83,84,85], reflecting the tumors’ potential to interfere with cardiac function depending on their size and location. Notably, the right atrium and right ventricle were the most commonly affected sites [10,11,14,15,16,20,21,22,23,29,32,34,35,36,37,40,41,43,44,46,47,48,49,50,51,52,55,56,57,58,77], which may hint at anatomical or hemodynamic factors influencing tumor development. Involvement of valvular structures, particularly the mitral valve [12,25,27,39,45,53,54], was also observed and may carry greater clinical significance due to the risk of functional impairment.

Tumor size varied widely, ranging from less than 1 cm [12,24,45,82] to over 11 cm [20,36], further supporting the notion that clinical impact is not dependent on histological type but rather on tumor size and anatomical context. This variability reinforces the importance of individualized clinical assessment, especially when determining the necessity and urgency of intervention.

Echocardiography remains the cornerstone of diagnostic work-up, used in over 80% of cases [10,11,12,13,16,22,23,24,25,26,27,28,29,30,32,34,35,36,37,38,39,40,41,42,43,44,45,46,47,48,50,51,52,53,54,55,56,57,81,82,84,85,86], followed by computed tomography [14,15,19,21,31,49,58] and, less frequently, chest X-ray [20,77,83]. Employing echocardiography as a first point of contact within the cardiology or general medical setting ensures that even subtle or asymptomatic lesions are not overlooked, thereby improving clinical outcomes through earlier detection, regardless of age. Given its accessibility, safety profile, and diagnostic yield, transthoracic echocardiography should be performed as an initial imaging modality in all patients with suspected cardiac pathology. This is particularly relevant in the context of cardiac hemangiomas, where early imaging can facilitate timely diagnosis and guide appropriate therapeutic intervention. However, CT continues to serve as a valuable adjunct [14,15,19,21,31,49,58,87] for anatomical delineation and surgical planning, providing high-resolution visualization of cardiac structures and tumor morphology.

Although none of the 55 cases included in this review reported the use of cardiac MRI as part of the diagnostic work-up, this modality holds considerable promise in the evaluation of cardiac hemangiomas. MRI offers high-resolution soft tissue contrast and can identify key features—such as T2-weighted hyperintensity and strong post-contrast enhancement—that reflect the vascular nature of these lesions, aiding in the differentiation of hemangiomas from other cardiac tumors. Nevertheless, conventional imaging alone often lacks the specificity required for definitive histological diagnosis, highlighting the potential value of more advanced imaging techniques, including cardiac MRI with tissue mapping, perfusion studies, or even positron emission tomography (PET), which may enhance diagnostic accuracy. Incorporating these modalities could facilitate more precise preoperative assessment, guide clinical decision-making, and potentially reduce the need for invasive diagnostic procedures when histopathological characteristics are strongly suggested by imaging profiles [80,88].

Management of cardiac hemangiomas is primarily surgical [2,10,11,12,13,14,15,16,19,20,21,22,23,24,25,26,27,28,29,30,31,32,33,34,35,36,37,38,39,40,41,42,43,44,45,46,47,48,49,50,51,52,53,54,55,56,57,58], with 87.3% of cases undergoing tumor excision. This approach is driven by the potential for complications such as embolism, arrhythmia, or obstruction [10,11,12,13,14,15,16]. Surgical excision was associated with a lower likelihood of post-treatment tumor reduction, not due to inefficacy but rather reflecting the complete removal of the lesion, which precludes further measurement of regression. This distinction highlights the curative potential of surgery in benign cardiac tumors. Conversely, non-surgical cases [46,77,82,84,86], although few, may represent patients with high surgical risk, asymptomatic profiles, or terminal illness.

Follow-up practices varied significantly across studies, with most cases reporting follow-up durations of less than 12 months [10,15,16,22,23,27,31,32,36,37,39,40,45,47,48,55,57,58,84]. Despite this heterogeneity, a moderate positive correlation (r = +0.51) was observed between follow-up duration and stable disease classification, suggesting that prolonged monitoring is associated with sustained disease stability, particularly in non-surgically managed or partially resected cases [81,83]. In contrast, shorter follow-up durations were noted in surgically treated patients [46,77,82,84,86], likely reflecting a lower perceived risk of recurrence post-excision. These findings underscore the need for more standardized follow-up protocols tailored to treatment modality and patient risk factors.

Co-occurrence with other tumors or systemic disorders was rare, although isolated cases of liver hemangiomas [15,82], malignancies [15,16,49,82,84], and hematologic abnormalities [16,28] were noted. These may represent coincidental findings; however, further research is warranted to explore potential syndromic associations or shared pathogenic pathways.

Our analysis affirms the rarity yet clinical importance of cardiac hemangiomas. The predominance of surgical management reflects the current clinical consensus favoring complete resection to prevent complications and achieve a definitive diagnosis. The variability in clinical presentation, anatomical distribution, and follow-up practices highlights the need for a multidisciplinary approach and further research to establish evidence-based management and surveillance guidelines. Future studies should aim to standardize diagnostic criteria, investigate molecular and genetic underpinnings, and assess long-term outcomes to improve care for patients with this uncommon but potentially impactful cardiac lesion.

A review of the literature reveals notable variability in the reported number of cardiac hemangioma cases over time, reflecting both the rarity of the condition and the evolution of diagnostic practices. For instance, McAllister and Fenoglio (1978) [89] identified only 15 cases, accounting for approximately 5% of all cardiac tumors described at the time, while Brizard et al. (1993) [3] reported a modest increase to 23 documented cases. In contrast, a more recent and extensive review by Li et al. (2015) [2] compiled 202 cases, suggesting a growing awareness and improved detection of these lesions. In the present review, we analyzed 55 cases published within the last five years (Appendix B), further emphasizing the upward trend in case identification. These discrepancies likely reflect advancements in imaging modalities, broader reporting standards, and increased clinical vigilance rather than inconsistencies in data collection and underscore the dynamic nature of knowledge surrounding this rare cardiac entity.

This review is subject to inherent limitations associated with its reliance on case reports and series. Potential publication bias likely skews the dataset toward unusual or successful outcomes, while incomplete reporting—particularly in follow-up duration and recurrence—impedes comprehensive evaluation. The absence of standardized diagnostic and therapeutic protocols across studies further contributes to data heterogeneity, limiting the generalizability of findings.

The observed dominance of echocardiography as the initial diagnostic modality underscores its critical role in the early detection of cardiac tumors. Its accessibility and sensitivity support its use as a frontline tool in cardiac mass evaluation. Furthermore, the high success rate of surgical resection and the absence of recurrence in most cases reinforce surgery as the primary therapeutic approach when feasible, particularly in symptomatic patients or tumors with embolic potential.

## 5. Conclusions

Cardiac hemangiomas are rare but clinically significant benign vascular tumors with a wide spectrum of presentations, anatomical locations, and histopathological subtypes. Despite the absence of a standardized classification system specifically for cardiac hemangiomas, current practice relies on subtyping into capillary, cavernous, mixed capillary/cavernous, and arteriovenous forms.

The findings from this review of 55 cases published in the past five years highlight a growing recognition and improved detection of cardiac hemangiomas, likely facilitated by advances in imaging techniques—particularly echocardiography and CT. These modalities remain essential for non-invasive diagnosis and preoperative planning, while surgery remains the cornerstone of management, offering curative outcomes in the majority of cases.

The heterogeneity in clinical presentation, follow-up duration, and demographic distribution—along with the scarcity of long-term data—underscores the need for greater standardization in diagnostic protocols and post-treatment surveillance. Furthermore, the limited number of cases reported in the literature continues to restrict large-scale statistical analyses, reinforcing the importance of systematic reporting and future multicenter collaborations.

### 5.1. Clinical Recommendations

**Imaging Strategy:** Transthoracic echocardiography (TTE) should be employed as the first-line diagnostic modality due to its accessibility, non-invasiveness, and high sensitivity for intracardiac masses. Advanced imaging techniques such as cardiac computed tomography (CT) or magnetic resonance imaging (MRI) should be reserved for cases where surgical planning is required or when echocardiographic findings are inconclusive.

**Surgical Indications:** Early surgical resection should be strongly considered for all symptomatic cardiac hemangiomas, particularly those with proximity to valvular structures or conduction pathways, due to the increased risk of hemodynamic compromise or arrhythmia. In contrast, asymptomatic or incidentally discovered tumors in patients with elevated surgical risk may be managed conservatively with ongoing clinical and imaging surveillance [90].


**Proposed Follow-Up Protocol:**
—Post-surgical patients: TTE should be performed at 6 and 12 months following surgery, with annual follow-up imaging thereafter for a duration of 2–3 years to monitor for recurrence or complications.—Non-surgical cases: MRI or TTE should be conducted every 6 to 12 months, with the interval determined by tumor size, growth dynamics, and anatomical considerations.—High-risk or symptomatic lesions under observation: More frequent imaging—every 3 to 6 months initially—may be warranted to detect early changes requiring intervention.


This stratified approach to diagnosis, treatment, and surveillance is intended to balance the benign nature of cardiac hemangiomas with their potential for functional disruption, thereby supporting individualized patient care based on risk assessment and tumor behavior. To assist clinicians in applying this approach, we propose a structured clinical protocol summarizing key decision points in evaluation, management, and follow-up (Figure 8).

### 5.2. Study Limitations and Future Directions

This systematic review is limited by its exclusive reliance on case reports and series, which introduces several sources of bias. First, publication bias likely favors unusual or favorable outcomes, limiting generalizability. Second, data heterogeneity—particularly in tumor size reporting, follow-up duration, and recurrence documentation—impedes consistent analysis. Third, the lack of standardized diagnostic, surgical, and follow-up protocols across studies complicates comparison and synthesis. Lastly, long-term outcomes remain underreported in many cases.

To address these challenges, we strongly advocate for the creation of multicenter prospective registries dedicated to cardiac hemangiomas. Such collaborations would support standardized data collection, reduce bias, and facilitate the development of evidence-based clinical pathways. In rare conditions such as this, coordinated efforts are essential to improve diagnostic precision, therapeutic outcomes, and surveillance practices across institutions.

## Figures and Tables

**Figure 1 cancers-17-01532-f001:**
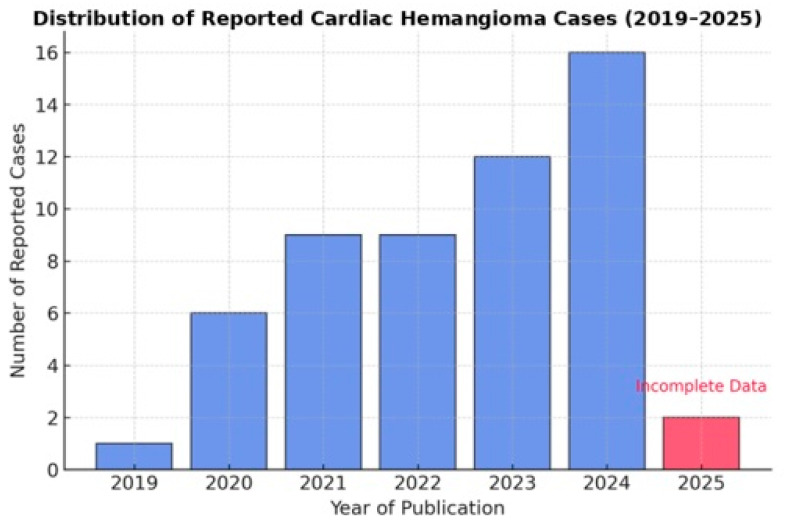
Temporal distribution of reported cardiac hemangioma cases (2019–2025).

**Figure 2 cancers-17-01532-f002:**
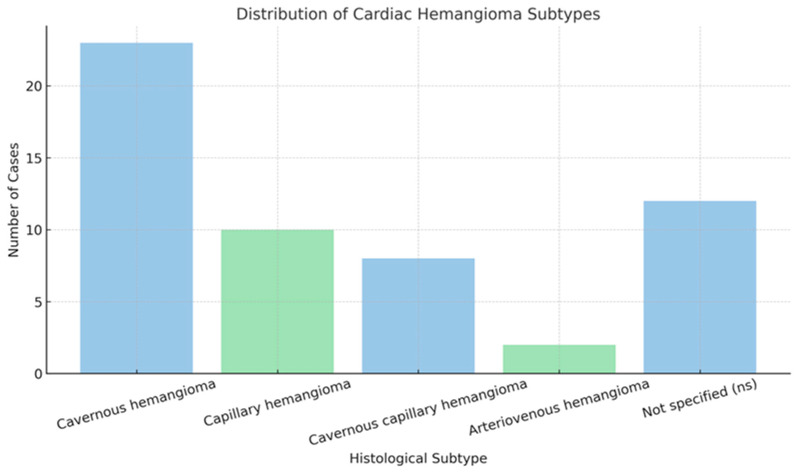
Histogram of hemangioma subtypes: distribution and frequency.

**Figure 3 cancers-17-01532-f003:**
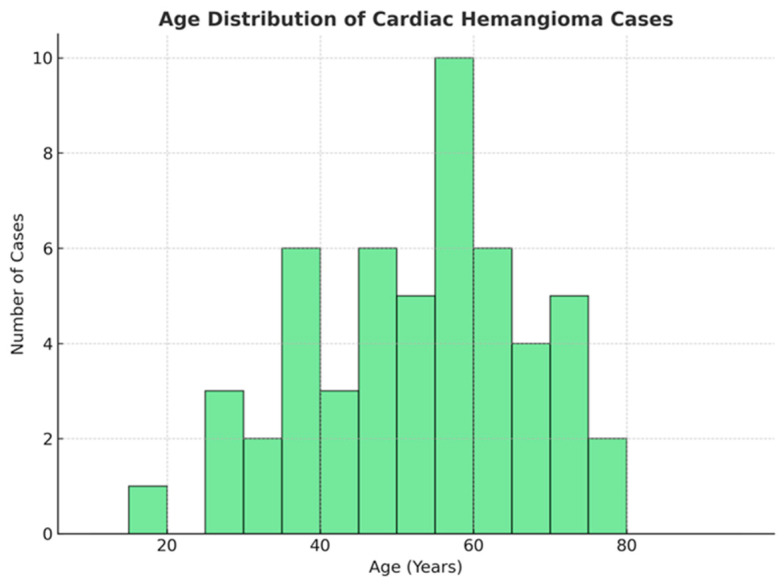
Age distribution of patients with cardiac hemangiomas.

**Figure 4 cancers-17-01532-f004:**
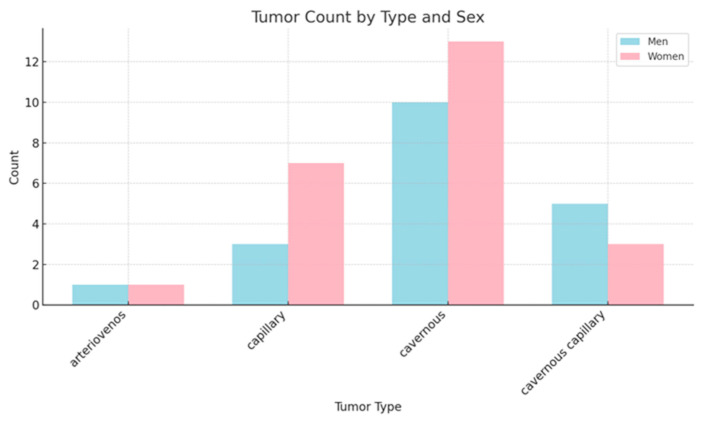
Analysis of distribution by type of tumor and gender.

**Figure 5 cancers-17-01532-f005:**
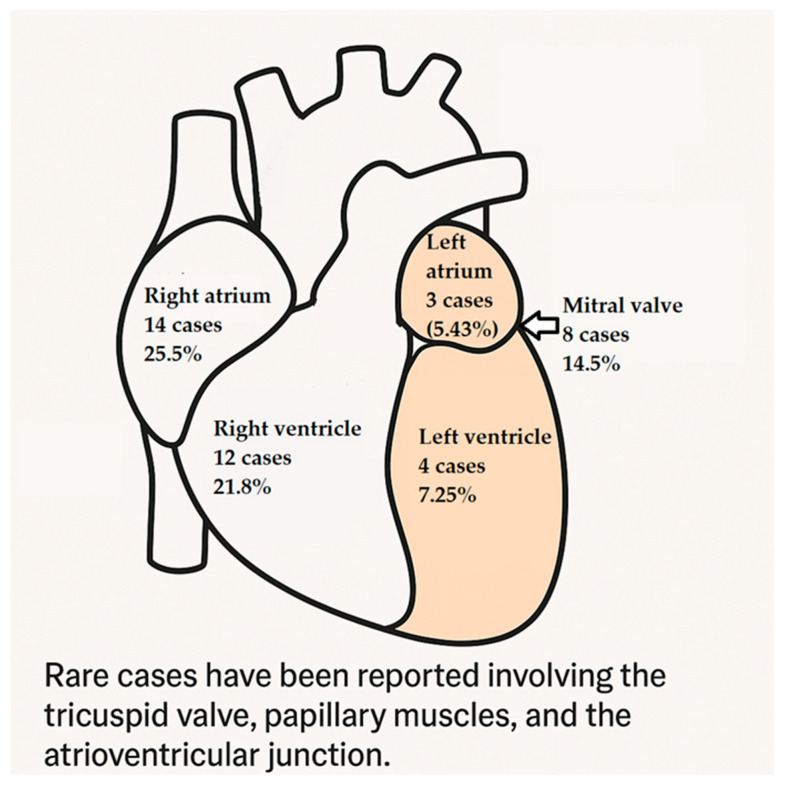
Anatomical distribution of cardiac hemangiomas by heart chamber.

**Figure 6 cancers-17-01532-f006:**
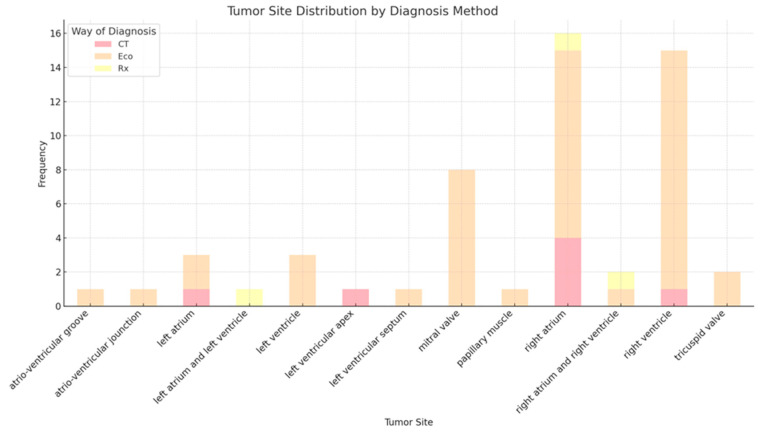
Tumor site distribution by way of diagnosis.

**Figure 7 cancers-17-01532-f007:**
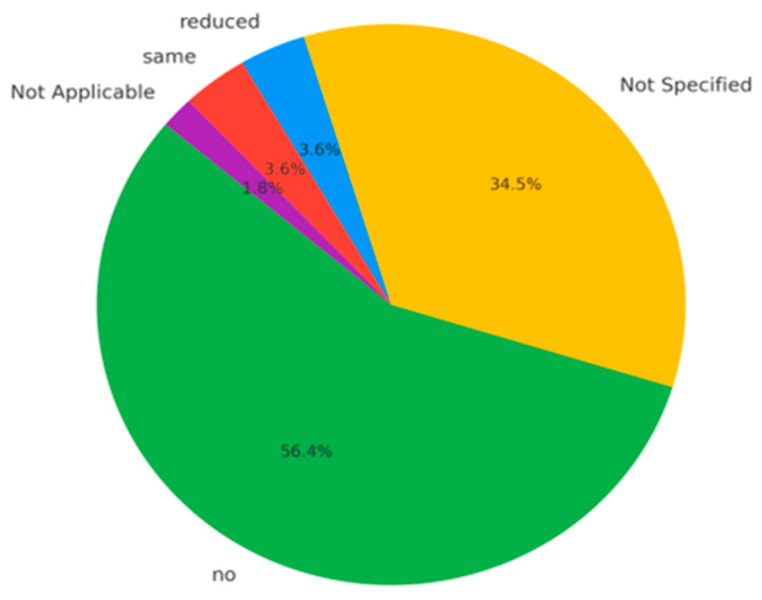
The outcomes of follow-up.

**Figure 8 cancers-17-01532-f008:**
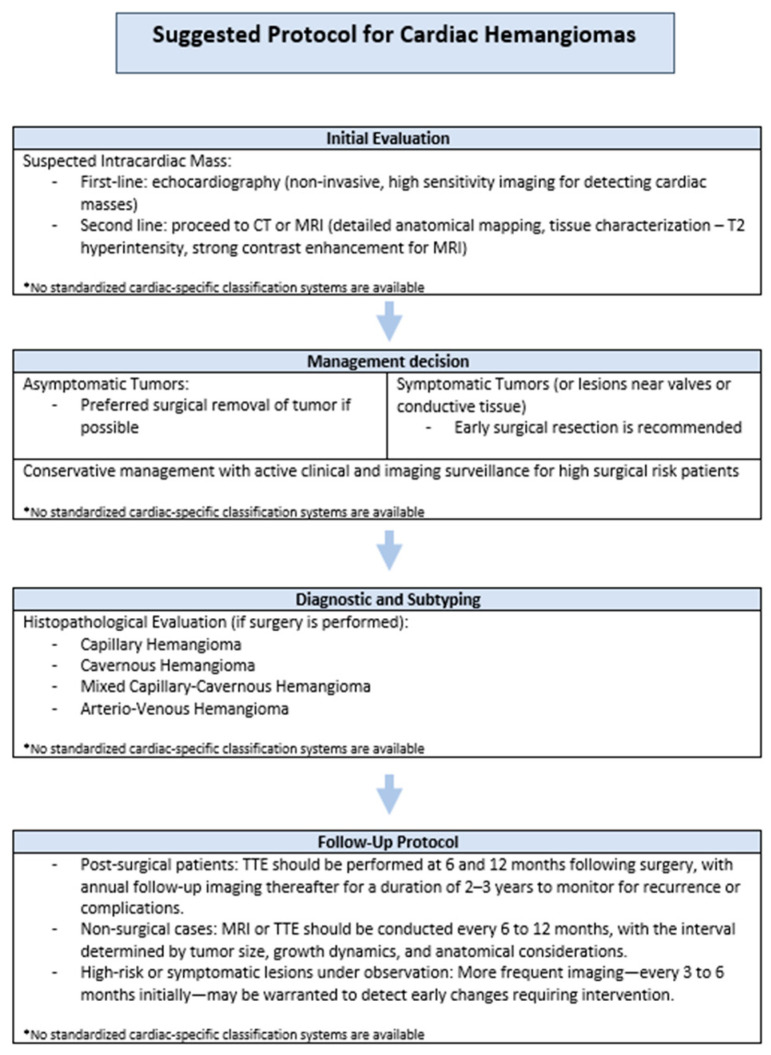
**Suggested protocol for cardiac hemangiomas.** The diagram outlines a proposed clinical approach to diagnosis, management, and follow-up based on findings from the current review and is intended to guide care in the absence of standardized protocols. The asterisk (*) indicates that no standardized classification or management protocols currently exist for cardiac hemangiomas. It serves as a reminder that the suggested steps are based on expert opinion and available literature, not formal consensus guidelines.

**Table 1 cancers-17-01532-t001:** Summary of age distribution by tumor type and gender.

Type	Avg_Age_m	Avg_Age_w	Count_m	Count_w	Max_Age_m	Max_Age_w	Min_Age_m	Min_Age_w
arteriovenous	52	55	1	1	52	55	52	55
capillary	68.66	56.85	3	7	78	71	52	48
cavernous	48.7	57.23	10	13	71	87	14	38
cavernous/capillary	60.6	45.33	5	3	79	52	14	34

## Data Availability

The original contributions presented in this study are included in the article/Appendix A. Further inquiries can be directed to the corresponding authors.

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
