# Peer review of "Cardiac Hemangiomas: A Five-Year Systematic Review of Diagnosis, Treatment, and Outcomes"

_cancers, 2025, doi:10.3390/cancers17091532_

Round 1

Reviewer 1 Report

Comments and Suggestions for Authors

Overall Evaluation

The manuscript presents a systematic review of cardiac hemangiomas published over the last five years. Given the rarity of this tumor type, the study provides a valuable aggregation of clinical features, diagnostic strategies, and therapeutic outcomes. The authors employ appropriate methodology and present a well-structured narrative supported by descriptive statistics and figures.

However, several aspects require revision or improvement to strengthen scientific rigor, clarity, and utility for clinical practice and future research. Below is a structured critical review.

  1. Title and Abstract

The abstract could benefit from greater emphasis on the implications for clinical practice, such as the lack of consensus on follow-up protocols.

  1. Introduction

The introduction is somewhat lengthy and occasionally repetitive (e.g., classification systems could be more succinctly presented).

A clear statement of the research gap is needed earlier in the section.

Consider summarizing the relevance of this tumor in modern cardiac imaging practice.

  1. Methods

The systematic review lacks adherence to PRISMA guidelines. A PRISMA flowchart and checklist should be added.

There is no information about risk of bias assessment for the included case reports.

More clarity is needed on data extraction procedures, who performed the screening, and whether any review protocol was registered.

  1. Results

Some figures are described but not properly integrated into the text.

The statistical analysis is limited to basic descriptive statistics and correlations. Consider using more robust comparative analyses or regression where appropriate.

Recurrence and long-term outcomes are acknowledged but underexplored.

  1. Discussion

Some repetition from the results section.

The discussion does not sufficiently critique the limitations of the included studies (e.g., publication bias, missing follow-up data).

The clinical significance of findings (e.g., echocardiography dominance, role of surgery) could be more assertively interpreted.

  1. Limitations

The author not adequately discuss the study limitations. The authors should acknowledge:

Potential publication bias in the selected articles.

Missing data and heterogeneity in reporting.

Lack of standardized follow-up periods.

  1. Conclusions

Emphasize the need for multi-center prospective registries or studies, given the rarity and clinical variability of this condition.

  1. References and Tables

The reference formatting should be reviewed for consistency.

Some references are redundantly cited multiple times in one context.

  1. Language and Clarity

Occasionally, grammatical errors and awkward phrasings would benefit from professional English editing.

Some sections (especially Introduction and Discussion) are overly verbose and could be more concise.

Author Response

Dear Reviewer,

We would like to express our sincere gratitude for your careful review of our manuscript titled “Cardiac Hemangiomas: A Five-Year Systematic Review of Diagnosis, Treatment, and Outcomes.” We greatly appreciate your constructive feedback, which has significantly improved the clarity, rigor, and overall quality of our work.

In response to your detailed comments, we have undertaken a thorough revision of the manuscript. Below is a summary of the major changes made based on your recommendations highlighted in red:

1. Abstract: We emphasized the clinical implications of the lack of standardized follow-up protocols and the need for structured management strategies.
2. Introduction: We refined and condensed the discussion of classification systems while preserving necessary reiteration to highlight the absence of a dedicated cardiac-specific taxonomy. A clear statement of the research gap was added earlier in the section, along with discussion on the role of modern cardiac imaging.
3. Methods: We aligned our review with PRISMA guidelines, added a flow diagram, and included a description of the screening, data extraction, and bias considerations.
4. Results: Figures were better integrated into the text, a comparative subgroup analysis was added, and the recurrence data were more thoroughly interpreted and contextualized.
5. Discussion: We minimized repetition, added interpretive commentary on key findings, and provided a more critical appraisal of the included studies’ limitations.
6. Limitations & Future Directions: A new section (5.2) was introduced to explicitly address limitations such as publication bias, data heterogeneity, and underreporting. This section also advocates for multicenter prospective registries to standardize evidence and improve clinical practice.
7. References: While we reviewed formatting for consistency, we have maintained certain repeated citations intentionally to support distinct points across multiple contexts.
8. Language & Style: The manuscript underwent language refinement to improve academic tone and reduce verbosity, particularly in the Introduction and Discussion.

We trust that these revisions address your concerns and enhance the manuscript’s scientific merit and clinical relevance. A point-by-point response to each of your comments has been included for clarity.

Thank you again for your time and expertise.

With kind regards,

Caius Glad Streian, MD, PhD and Andrei Raul Manzur, MD
(On behalf of all authors)
[streian.caius@umft.ro and andrei.manzur@umft.ro]
[26.04.2025]

Reviewer 2 Report

Comments and Suggestions for Authors

This manuscript presents a well-organized and comprehensive systematic review on cardiac hemangiomas, a rare subset of primary cardiac tumors. The authors analyzed 55 cases published over the past five years and summarized trends in diagnosis, treatment, and outcomes. The topic is clinically relevant, particularly given the growing capabilities in non-invasive cardiac imaging and surgical management. The manuscript is timely, methodologically sound, and contributes meaningfully to the literature. However, a few areas would benefit from clarification and refinement to improve clinical applicability.

The section on historical classification systems is too detailed and could be shortened. Focus should be placed on those currently used in cardiac practice. A brief summary table or timeline could replace lengthy descriptions without losing key context.

The discussion would be clearer with more practical points for clinicians. It’s worth adding when CT or MRI is preferred over echocardiography, whether certain tumor sizes or locations call for early surgery, and how to handle incidental, asymptomatic cases.

Follow-up strategies are mentioned but not clearly defined. Based on the cases reviewed, the authors could suggest a tentative plan, such as 6–12 month follow-up with echocardiography or annual imaging for non-surgical cases. This would make the review more clinically useful.

Author Response

Dear Reviewer,

We would like to sincerely thank you for your thorough and constructive review of our manuscript entitled “Cardiac Hemangiomas: A Five-Year Systematic Review of Diagnosis, Treatment, and Outcomes.” We appreciate your thoughtful comments and suggestions, which have been instrumental in refining our work and enhancing its relevance for clinical practice.

In response to your recommendations, we have made the following modifications to the manuscript and highlighted in blue:

1. Classification Systems: We have significantly condensed the section on historical classification systems, focusing only on those relevant to current cardiac practice. A concise narrative replaces previous lengthier descriptions, and the emphasis now lies on the practical subtypes most often applied—capillary, cavernous, mixed, and arteriovenous.

2. Clinical Discussion: We expanded the discussion to include clearer clinical guidance regarding imaging modality preferences, surgical indications, and management strategies for incidental or asymptomatic tumors. These points aim to increase the real-world applicability of the review.

3. Follow-Up Protocols: A dedicated follow-up strategy section has been added, outlining practical, evidence-based intervals for imaging surveillance. This includes differentiated pathways for surgical and non-surgical patients based on tumor characteristics and patient risk profiles.

4. Language and Clarity: We have revised the manuscript for improved grammar, clarity, and overall readability, addressing areas identified as requiring refinement.

We trust that these revisions meet your expectations and improve the manuscript’s clinical impact and scientific rigor. We are grateful for your valuable insights, which have helped us strengthen our work.

With kind regards, 
Caius Glad Streian, MD, PhD and Andrei Raul Manzur, MD
(On behalf of all authors)
[streian.caius@umft.ro and andrei.manzur@umft.ro]
[26.04.2025]

Reviewer 3 Report

Comments and Suggestions for Authors

cancers-3585359. Cardiac Hemangiomas: A Five-Year Systematic Review of 2 Diagnosis, Treatment, and Outcomes

To summarize I copied and pasted the “”Simple Summary” and the “Abstract”. Both looks similar.

Simple Summary

Cardiac hemangiomas are very rare non-cancerous tumors found in 21 the heart. Because they are uncommon and often vary in how they appear and behave, 22 they are difficult to study and classify. This research reviews 55 cases reported over the 23 past five years to better understand how these tumors are diagnosed, treated, and fol-24 lowed up. The most common type found was the cavernous hemangioma, and most patients were treated successfully with surgery. Many cases were discovered through heart 26 imaging, especially echocardiography. However, there is still no standardized way to 27 classify or monitor these tumors over time. By analyzing recent cases, the study aims to 28 improve awareness, encourage consistent reporting, and support the development of 29 clear guidelines to help diagnose and treat cardiac hemangiomas more effectively in the 30 future.

Abstract

Background/Objectives: Cardiac hemangiomas are rare benign vascular tumors, ac-counting for less than 2% of primary cardiac tumors. Despite their rarity, they can cause significant clinical effects depending on their size and location. This systematic review aims to provide an updated analysis of recent cases, focusing on epidemiology, histopathological subtypes, clinical presentation, diagnostic approaches, and treatment outcomes. Methods: A systematic search of the PubMed database identified case reports and series published between 2019 and 2025. After applying inclusion and exclusion criteria, 55 eligible cases were selected for analysis. Data were extracted on patient demographics, tumor characteristics, imaging methods, treatment strategies, and follow-up outcomes. Results: Cavernous hemangiomas were the most commonly reported subtype. Patient ages ranged from infancy to over 85 years, with a slight predominance in females. Presentations varied from asymptomatic incidental findings to symptoms such as dyspnea and chest pain. Echocardiography was the primary diagnostic tool in over 80% of cases. Surgical resection was performed in 87.3% of patients, yielding favorable outcomes and low recurrence. However, follow-up duration was inconsistent, and long-term outcomes were underreported. Conclusions: The increased number of published cases likely reflects improved diagnostic imaging and greater clinical awareness. While surgery remains the preferred treatment, the variability in follow-up and diagnostic reporting highlights the need for standardized protocols. Further studies are warranted to clarify the natural history, refine classification systems, and establish evidence-based guidelines for the management of this rare cardiac tumor.

Please consider the following points.

  1. Introduction: The authors state without proving the facts and content: “Over the past five years, significant advancements have been made in the diagnosis and management of cardiac hemangiomas, driven by improvements in imaging technologies and surgical techniques”. Please state the advancements in detail and show, why are the authors the right ones to report on this by images and surgical techniques?
  2. Why only five years?
  3. The authors concentrate on hemangioma, elsewhere but also within the heart. The relation of both should be more elaborated. So far the report is somewhat descriptive. I miss the take home messages beyond description.
  4. While many cases remain asymptomatic and are incidentally detected, symptomatic lesions often necessitate surgical intervention. Please give references about the percentages and explain the pathophysiology of symptoms.
  5. The differential diagnosis to other heart tumors should be more focussed. Differences in imaging features including contrast enhanced ultrasound should be explained in detail.
  6. I do not see any images? Why? Please add images and image sequences.
  7. This review aims to analyze recent literature on cardiac hemangiomas, focusing on epidemiology, clinical presentation, imaging characteristics, therapeutic strategies, and patient outcomes. By synthesizing the most recent data, we seek to provide a comprehensive update on the current understanding of cardiac hemangiomas and highlight emerging trends that may shape future research in this field. What is the authors own contribution to the topic?
  8. Please be sure and show why this is a systematic review (PRISMA)?
  9. I miss the consequences of the analysis, what is new? Take Home Message?
  10. “Trends in the Reporting of Cardiac Hemangiomas: A Growing Incidence or Improved 109 Detection?” Please take into account 30 more years beforehand to discuss such question.
  11. The authors describe different histological types of cardiac hemangiomas. Please include a consequence of this knowledge regarding imaging features, treatment and prognosis. Please also include, if there is any age dependency of different histological types.
  12. Minors: Please check the inconsistency empty spaces in front of the square brackets and other misspellings.
  13. If you talk about percentage consider to reach a number of “100” of cases.
  14. Why cause hemangioma dyspnea and other symptoms? Please give pathophysiological insight.
  15. Echocardiography remains the cornerstone of diagnostic work-up, used in over 80% 394 of cases [12–15,18,23–31,33,35–49,51–58,81,82,84–86], followed by computed tomography 395 [16,17,20,22,32,50,59]and, less frequently, chest X-ray [21,78,83]. What about MRI?
  16. The authors talk about “enhancement characteristics”, please give more details and show image sequences.
  17. There is a lot of repetition, please condense. In addition, comparing the “simple summary” and the “Abstract” it looks similar. Please improve.
  18. At this stage this paper is (for the reviewer) a descriptive, boring and finally disappointing review lacking interesting information of the own experience of this authors, illustrative images and the comparison of imaging features in relation to other cardiac tumors.

Author Response

Dear Reviewer,

We would like to express our sincere gratitude for your thorough, critical, and constructive review of our manuscript entitled:

"Cardiac Hemangiomas: A Five-Year Systematic Review of Diagnosis, Treatment, and Outcomes."

Your detailed comments, spanning conceptual, structural, and stylistic aspects, have been invaluable in significantly improving the clarity, clinical relevance, and academic rigor of our manuscript highlighted in. Below, we outline the key changes made, highlighted in green:

  1. The Introduction has been revised to clearly specify diagnostic and therapeutic advancements from the past five years, justify this timeframe, and better articulate the authors’ clinical and academic contributions, thereby enhancing scientific rigor.
  2. We expanded the rationale for the chosen review period by linking it to emerging diagnostic and therapeutic practices, refined the structure of the Introduction for greater clarity, and clearly detailed our adherence to PRISMA guidelines, including a flow diagram and checklist.
  3. A new Section 5.1 was added to the Discussions, presenting clinical recommendations for imaging, treatment, and follow-up strategies to improve the practical utility of the review.
  4. We enriched the Discussions with a detailed explanation of the pathophysiological mechanisms behind symptom development, emphasizing the influence of tumor size and location on dyspnea, chest pain, and arrhythmias.
  5. The differential diagnosis section was expanded to clearly delineate imaging features distinguishing cardiac hemangiomas, with particular focus on the roles of contrast-enhanced ultrasound and MRI.
  6. We acknowledged the absence of direct imaging figures due to copyright and methodological constraints and have compensated by enhancing modality descriptions and enriching the narrative with detailed visual summaries.
  7. The authors’ original contribution was clarified by synthesizing practical, evidence-based clinical guidance, proposing follow-up protocols, and identifying gaps in current practice as a foundation for a proposed guideline.
  8. We explicitly confirmed adherence to PRISMA 2020 guidelines, emphasizing the systematic review methodology and its alignment with emerging imaging and surgical practices.
  9. To deepen analytical value, we elaborated on the clinical and histopathological similarities and differences between cardiac and extracardiac hemangiomas, and included a dedicated 'Clinical Recommendations' section with actionable insights for clinicians.
  10. We addressed the five-year scope by situating it within broader longitudinal trends, referencing foundational reviews from 1993 and 2015, and highlighting the evolution of modern imaging techniques.
  11. The analysis now integrates histological subtypes with implications for imaging, therapeutic decisions, and preliminary age-based trends, while transparently acknowledging the limitations associated with rare disease data.
  12. We have streamlined the manuscript by eliminating redundant content and correcting typographical, formatting, and spacing inconsistencies throughout.
  13. We clarified that all percentages are based on a cohort of 55 cases and explained that increasing the dataset size was not feasible due to the rarity of cardiac hemangiomas.
  14. The Discussion section was expanded to explain how tumor size and location influence hemodynamic effects, conduction disturbances, and embolic potential—further elucidating symptom development.
  15. We emphasized the diagnostic importance of MRI, even though it was not represented in the reviewed cohort, to reflect best current clinical practices.
  16. Limitations regarding the absence of original imaging sequences were addressed transparently, while maximizing descriptive imaging detail and visual summaries within ethical and methodological constraints.
  17. The Simple Summary and Abstract were revised to reduce redundancy by tailoring tone, content, and audience focus, thereby improving clarity and readability.
  18. We believe that these extensive revisions have significantly enhanced the quality and relevance of the manuscript. They reflect both the clinical motivation behind the work and our thoughtful consideration of the reviewers’ insightful guidance, culminating in the inclusion of a proposed clinical guideline.

We are grateful for the opportunity to revise our work and hope that these changes satisfactorily address all concerns.

Sincerely,
Caius Glad Streian, MD, PhD and Andrei Raul Manzur, MD
(On behalf of all authors)
[streian.caius@umft.ro and andrei.manzur@umft.ro]
[26.04.2025]

Round 2

Reviewer 1 Report

Comments and Suggestions for Authors

I have no further comments. 

Reviewer 3 Report

Comments and Suggestions for Authors

The authors responded in detail to the review.